# The Multifaceted Role of Annexin A1 in Viral Infections

**DOI:** 10.3390/cells12081131

**Published:** 2023-04-11

**Authors:** Filipe Resende, Simone de Araújo, Luciana Pádua Tavares, Mauro Martins Teixeira, Vivian Vasconcelos Costa

**Affiliations:** 1Post-Graduation Program of Cell Biology, Department of Morphology, Biological Sciences Institute, Federal University of Minas Gerais, Belo Horizonte 31270-901, Brazil; 2Center for Research and Development of Drugs, Biological Sciences Institute, Federal University of Minas Gerais, Belo Horizonte 31270-901, Brazil; 3Pulmonary and Critical Care Medicine Division, Department of Medicine, Brigham and Women’s Hospital and Harvard Medical School, Boston, MA 02115, USA; lpaduatavares@bwh.harvard.edu; 4Department of Biochemistry and Immunology, Biological Sciences Institute, Federal University of Minas Gerais, Belo Horizonte 31270-901, Brazil

**Keywords:** Annexin A1, FPR2, virus, infection, inflammation, host-directed therapies

## Abstract

Dysregulated inflammatory responses are often correlated with disease severity during viral infections. Annexin A1 (AnxA1) is an endogenous pro-resolving protein that timely regulates inflammation by activating signaling pathways that culminate with the termination of response, clearance of pathogen and restoration of tissue homeostasis. Harnessing the pro-resolution actions of AnxA1 holds promise as a therapeutic strategy to control the severity of the clinical presentation of viral infections. In contrast, AnxA1 signaling might also be hijacked by viruses to promote pathogen survival and replication. Therefore, the role of AnxA1 during viral infections is complex and dynamic. In this review, we provide an in-depth view of the role of AnxA1 during viral infections, from pre-clinical to clinical studies. In addition, this review discusses the therapeutic potential for AnxA1 and AnxA1 mimetics in treating viral infections.

## 1. Introduction

The COVID-19 (Coronavirus Disease 19) pandemic has brought great attention to the undeniable threat of emerging and re-emerging viral diseases to public health worldwide. According to the COVID-19 Excess Mortality Collaborators, over 18.2 million people died due to COVID-19 during the period between 1 January 2020 and 31 December 2021 [1]. In addition, several other epidemics and pandemics, with viruses as causative pathogens, have led to increased human and economic losses every year. Ebola, Zika, Influenza A, Chikungunya and Dengue virus are examples of relevant viruses causing severe disease in humans. For instance, approximately 294,000 to 518,000 deaths are associated to influenza infections annually [2,3,4]. In addition, more than 150 enteric viruses are associated with waterborne human disease [5], making viral infections a substantial challenge to public health. Given the recent reports of viral disease outbreaks around the world, a significant effort has been made for the development of effective vaccines and antiviral drugs [6,7]. The development of novel treatment strategies is urgently needed.

Inflammation is a highly coordinated physiological response triggered during infections to restrain pathogen proliferation and induce tissue repair. Once an insult (i.e., pathogen or injury) is neutralized, the ensuing response must come to an end to prevent extensive tissue damage or chronic inflammation [8]. Moreover, during certain viral infections, exaggerated inflammation appears to contribute to increased morbidity and mortality, as clearly exemplified by the beneficial effects of glucocorticoids in COVID-19 [9,10]. Therefore, different studies have attempted to identify novel immunomodulatory strategies to counter-regulate inflammation and reduce pathogenesis during infections. In contrast to the pathogen-directed therapeutics, treatment strategies focused on the host are less passive to resistance and can be useful for different types of pathogens that cause tissue damage and disease through similar immunopathologic mechanisms.

The termination of the inflammatory response has been revisited in the last decades as more than merely a passive and catabolic process. An active and dynamically regulated response initiates in the tissue during inflammation to induce the resolution of the inflammatory process [11,12]. Resolution of inflammation is mediated by the local production of specific molecules, collectively named pro-resolving mediators. The pro-resolving molecules include lipids called specialized pro-resolving mediators (SPMs), short-chain fatty acids (e.g., acetate) and proteins, including Annexin A1 (AnxA1) [13,14,15,16].

AnxA1 was shown to promote (1) the apoptosis of granulocytes, (2) efferocytosis of apoptotic cells, (3) phagocytosis of pathogens and debris, (4) induce the recruitment and polarization of regulatory macrophages, (5) inhibit recruitment of granulocytes and production of pro-inflammatory cytokines in the context of several inflammatory diseases [13,16,17]. In this regard, one would assume that AnxA1 is mainly protective in the context of viral diseases. However, recent studies have shown that the role of this molecule during viral infections is more complex than previously thought and it can be beneficial or detrimental depending on the pathogen and timing of production/administration [18,19,20,21].

Throughout previous years, our group has been exploring the idea of dissociating the components of the host immune system that can fight pathogens, from the components that cause excessive inflammation, which can be life-threatening [22,23,24,25,26]. This dissociation has, as one of its central players, the modulation of AnxA1 signaling and its membrane receptor, formyl Peptide Receptor 2 (FPR2). This signaling axis is a promising target for developing therapeutic strategies for viral infections. To this end, this review aims to briefly summarize the role of AnxA1 in the context of viral infections, from murine models to clinical investigations, focusing on the role of AnxA1 in respiratory and non-respiratory viral infections.

## 2. Inflammation, the Resolution Phase and Its Importance during Infections

Inflammation is a protective response of the immune system to harmful agents, such as those from infectious, traumatic, autoimmune or tumor sources [8]. This response results in the release of inflammatory mediators (e.g., histamine, prostaglandins, bradykinin) that act on blood vessels, increasing their permeability, causing vasodilation, and facilitate the migration of immune cells to the affected tissues. These events lead to the five cardinal signs of inflammation: redness, heat, swelling, pain and/or loss of function [27].

Acute inflammation begins immediately, lasts for a few days, and is mainly characterized by vascular events, with a predominance of neutrophils and an intense production of cytokines, such as IL-1β and IL-6 [8]. Most commonly, acute inflammation resolves itself and homeostasis is restored. However, if it does not, it can lead to tissue damage or progress to a chronic phase, characterized by the persistence of inflammation that can last for months to years. In chronic inflammation, there is a predominance of macrophages, fibroblasts and an intense formation of fibrotic tissue [28]. Thus, it is of paramount importance to treat acute inflammation properly to limit tissue damage associated with the acute response, and to prevent its progression to the chronic phase.

Briefly, during acute inflammation, endogenous proteins that act as sensors of pathogen-associated molecular patterns (PAMPs) activate pro-inflammatory signaling pathways, especially in leukocytes, triggering the inflammatory response. The recruitment of leukocytes to the affected tissue site and the production of specific mediators are common features of the inflammatory process, which may or may not result in the impairment of the primary functions of the affected organ [29,30].

During viral infections, cytosolic and membrane sensors expressed by different immune and non-immune cells recognize components of virus particles during the steps of infection and replication [31,32]. For instance, Toll-like receptor (TLR)-7 recognizes single-stranded viral RNAs, while TLR-3 and retinoic-acid-inducible gene 1 (RIG-1) are sensors for viral double-stranded RNA. This recognition stimulates specific transcription factors that lead to the activation of pro-inflammatory genes and the migration of immune cells to the inflammatory site [33].

The successful resolution of inflammation in the affected site depends on a wide variety of factors, including the clearance of pathogens, the switch from pro-inflammatory molecules to anti-inflammatory mediators, the inhibition of neutrophil recruitment, and the non-phlogistic recruitment of macrophages [34]. This is essential to promote pathogen clearance and reestablish the primary functions of the injured tissue without a significant loss of function and fibrosis formation [11].

The resolution phase can be traditionally classified by several events, such as the decrease in the production of proinflammatory cytokines and chemokines, inhibition of neutrophil recruitment, induction of neutrophil apoptosis, ceasing of survival signals on neutrophils, phagocytosis of apoptotic bodies by macrophages (prompting a phenotypic change from M1 to M2 macrophages), recruitment of non-phlogistic monocytes and production of SPMs [11,35,36,37]. Lipid components, such as Lipoxin A4, D- and E- series of Resolvins, Maresins, and Protectins are examples of SPMs [15,38]. Some proteins can also act as pro-resolving molecules, such as the proteins Angio (1–7) and AnxA1. In this sense, AnxA1 plays an important role in controlling the inflammatory response induced by viruses [39].

## 3. Annexin A1

AnxA1 was first described by Flower and Blackwell in 1979 as a second messenger that affected the production of prostaglandins [40]. Subsequently, AnxA1 was characterized as a protein that inhibited the activity of phospholipase A2 (PLA2) and was involved in the anti-inflammatory actions of glucocorticoids [41]. AnxA1 belongs to the superfamily of annexins that bind to membrane phospholipids in the presence of calcium ion (Ca^2+^). Most of its functions are related to its ability to interact with cell membranes. This interaction is reversible and regulated mainly by post-translational modifications, such as the phosphorylation of serine, tyrosine and histidine residues [42].

AnxA1 is expressed in a variety of immune cells, including neutrophils, monocytes/macrophages and mast cells, with a low expression on a subset of lymphocytes, such as T cells. Polymorphonuclear leukocytes (PMNs) contain large amounts of AnxA1, which can represent up to 4% of the total cytosolic proteins [43]. After PMN activation or adhesion to endothelial cell monolayers, AnxA1 is mobilized, externalized on the membrane and subsequently released into the extracellular medium [44,45,46].

The generation of AnxA1-knockout (AnxA1-KO) mice has allowed for multiple studies to better characterize the role of this protein in different contexts of inflammation [47,48,49]. AnxA1-KO mice demonstrate a prolonged and amplified inflammatory response after local stimulation (e.g., paw edema model), an increased extent of joint damage in the antigen-induced arthritis model and a greater susceptibility to nociception [50,51,52]. Additionally, AnxA1-KO neutrophils display increased endothelial transmigration and a heightened responsiveness to inflammatory stimuli, further demonstrating the importance of endogenous AnxA1 in the resolution of inflammation [53].

Interestingly, the role of AnxA1 varies depending on the inflammatory context. For instance, in some conditions, the full-length AnxA1 (37 kDa) can be cleaved to the isoform (33 kDa) by neutrophil proteases [54,55,56,57], and these peptides have been found in the lungs of mice in an experimental model of asthma [58], and in the bronchoalveolar fluid of patients with cystic fibrosis [59,60]. This suggests that the cleavage of AnxA1 may be related to a proinflammatory phenotype or inactivation of the protein. In other words, the cleavage of AnxA1 may dampen the anti-inflammatory effects of the intact protein in addition to the effect of some of these peptides, to promote a pro-inflammatory effect [61,62,63].

Neutrophils release specific proteases in a temporospatial context at the sites of inflammation. In mouse models in which neutrophils are the most prominent infiltrated leukocytes, a pattern of cleavage of AnxA1 is most clearly observed over the course of inflammation [25,64,65,66]. The N-terminal fragment of AnxA1, comprising the amino acid sequence 2 to 26 (Ac2-26), is one of the most common agonists used in experimental models to mimic some of the pro-resolutive effects of AnxA1 [13,16].

Curiously, while AnxA1 activates the FPR2 receptor, Ac2-26 can activate all three isoforms of FPR, leading to anti-inflammatory/pro-resolving effects, such as neutrophil apoptosis and macrophage efferocytosis [13,67,68], or pro-inflammatory responses such as increased expression of pro-inflammatory cytokines, for example, IL-1β, IL-6 and TNF-α [68]. An increased release of pro-inflammatory cytokines by translocation of AnxA1 to the cell nucleus has been observed in BV-2 microglial cells after oxygen glucose deprivation/reoxygenation injury [69]. Taken together, these studies highlight the complex biology of AnxA1 in the inflammatory microenvironment.

### 3.1. The AnxA1 Receptor

To employ its pro-resolutive actions, AnxA1 binds to a G-protein-coupled receptor called formyl peptide receptor 2 (FPR2). FPR2 is a member of the formyl peptide receptor family and is expressed by different leukocytes, such as neutrophils, macrophages and mast cells [70,71,72,73]. In addition to FPR2, humans express two other receptors, named FPR1 and FPR3. Beyond the three human counterparts, mice also express four different genes that encode chemoreceptors in the vomeronasal organ and sense pro-inflammatory molecules in the neuroepithelium, as elegantly demonstrated by Rivière and co-workers [74].

FPR2 binds to a plethora of compounds, from lipid molecules to peptides and proteins, some of which can exert pro-resolutive or proinflammatory actions through the receptor. Examples include LxA4, AnxA1, Resolvin D1 and Resolvin D3 as pro-resolutive, and the acute-phase protein serum amyloid A (SAA), LL-37, and the amyloid beta 42 peptide (Aβ-42) as pro-inflammatory compounds [75]. The chemical promiscuity of FPR2 and its versatility concerning the activation/inhibition of different signaling pathways is, at least partially, attributed to its unique extracellular structure, with a larger binding pocket that differs from other FPR receptors and its ability to homo- or heterodimerize with FPR1 and FPR3 [75].

In the last ten years, studies regarding the extracellular structure of the receptor have elucidated the fundamental differences between FPR1 and FPR2 in terms of affinity to diverse agonists. As extensively reviewed by Qin and colleagues, 2022, several characteristics allow for FPR2 to bind to different molecules. First, its larger binding pocket at the extracellular loop allows it to accommodate larger peptides compared with FPR1. Second, the charge distribution of the residues within the binding cavity differs from the two receptors, and cryo-EM structure studies combined with mutational in vitro assays revealed that transmembrane (TM) and different extracellular loop (ECL) domains of FPR2 participate in the binding of different agonists [76,77,78,79,80]. Furthermore, evidence points toward a role of aspirin-triggered 15-epi-lipoxin A4 (ATL) allosteric binding sites to modulate the activation of the receptor in the concomitant presence of other agonists [81,82]. As aforementioned, the homo- and heterodimerization of the receptor is mediated by different agonists and allows for the receptors to activate different signaling pathways. AnxA1 and LxA4 preferentially induce the homodimerization of FPR2, which in turn leads to the production of IL-10 via the p38/MAPKAPK/HSP27 pathway [83]. Nevertheless, AnxA1 and its mimetic peptide, Ac2-26, are also capable of inducing the heterodimerization of FPR2/FPR1, which leads to neutrophil apoptosis and may reduce the amount of monomeric FPR1 available for pro-inflammatory agonists [75].

The activation of AnxA1/FPR2 axis is related to diverse signaling pathways that have a direct or indirect impact on virus replication, such as extracellular signal-regulated kinase (ERK), and signal transducer and activator of transcription 3 (STAT3) [84,85,86,87,88]. Additionally, some viruses can hijack the binding of AnxA1 to the FPR2 receptor, and enhance the infection of the cells. Therefore, in the next sections, we will discuss what is currently known about the role of AnxA1 during viral infections (summarized in Table 1).

### 3.2. Annexin A1 during Respiratory Viral Infections

#### 3.2.1. SARS-CoV-2 (COVID-2019)

Exuberant inflammation is a poor prognostic marker for severe acute respiratory syndrome caused by SARS-coronavirus 2 (SARS-CoV-2) [103]. Indeed, persistent or unresolved inflammation can increase the severity of the disease, leading to organ failure, and ultimately death [104]. Excessive production of cytokines, known as a cytokine storm, has been observed in COVID-19 patients who needed treatment at the intensive care unit (ICU) upon hospital admission [105,106,107]. Therefore, excessive inflammatory responses in disease may be a predictor for complications, such as acute respiratory distress syndrome, multi-organ failure and death [108]. Interestingly, treatment with glucocorticoids has been shown to result in lower mortality in patients hospitalized with COVID-19 [109,110]. Ac2-26 peptide has been shown to reduce inflammation-induced thrombosis [111] and infectious pneumonia [89], both features of COVID-19. Therefore, AnxA1-based peptides or FPR2 agonists might hold great promise as therapeutic agents against COVID-19.

Until the end of 2022, few studies regarding AnxA1 and SARS-CoV-2 have been published. Canacik and colleagues identified AnxA1 as a biomarker for the severity of COVID-19, as the levels of the protein in the serum of patients with the severe disease were decreased compared to those of healthy volunteers, or even patients with mild disease [89]. However, another independent study from Ural and colleagues showed the opposite: the levels of AnxA1 in the serum of COVID-19 patients were increased compared to the control group. Importantly, in the latter study, COVID-19 patients were stratified in mild, moderate and severe groups, using thoracic computed tomography scans to evaluate the ground glass opacities and consolidation in the lungs [112].

Both studies collected blood from the patients upon hospital admission, as outlined in the manuscript methodology. Additionally, the studies used the same commercially available kit to measure the levels of AnxA1. Despite the similar methodology, the results of the two studies are contrasting, and this might be explained by the differences in the patient cohort at the time of hospital admission, which can be highly variable and thus reflect the levels of different markers of inflammation, such as AnxA1. In line with the finding of Ural and co-workers, another recently published study demonstrated that AnxA1 levels are augmented during COVID-19, and may be related to increased ICU admission [90]. A pre-print by Hoffmann and colleagues demonstrated that convalescent patients of COVID-19 have circulating monocytes that up-regulate the expression of AnxA1, suggesting that the levels of AnxA1 may be related to the convalescent state of the patients [113]. Although it has been suggested that AnxA1 analogues may be helpful in COVID-19 patients [114], it is difficult to reconcile the findings of all studies evaluating the concentration of AnxA1 in COVID-19 patients. Indeed, most, but not all, studies suggest that elevated levels of AnxA1 associate with disease severity and the convalescent state. It appears, therefore, that AnxA1 levels elevate in an attempt to prevent disease in highly affected patients, and may be relevant to the ensuing convalescence. It has not been shown beyond doubt whether this can be exploited therapeutically.

#### 3.2.2. Influenza Virus (IAV)

The Influenza virus (IAV) is known to cause major epidemics among the human population [115]. Different studies explored the role of AnxA1 during Influenza virus (IAV) infection. It is worth mentioning the groups of Lina Lim and Béatrice Riteau, which have previously independently demonstrated the importance of AnxA1/FPR2 axis for IAV infection. During IAV infection, AnxA1-deficient mice have a survival advantage when compared to wild-type (WT) animals, and this was associated to diminished viral loads in the lungs of AnxA1KO mice [85]. This was attributed to the increased replication of IAV in the presence of AnxA1 and the activation of FPF2. Interestingly, despite the survival advantage and the decreased viral titers, the absence of AnxA1 led to increased lung damage and infiltration of leukocytes [91]. Of those, augmented levels of neutrophils were found, which can be explained by the known role of AnxA1 in the adhesion and transmigration of these cells to the sites of inflammation [91,116]. Interestingly, a study by Schloer and co-workers showed that the treatment with human recombinant AnxA1, four days prior to infection of mice, expands the population of alveolar macrophages (AMs) and provides protection against infection [19].

At first, these results may seem conflicting, but it demonstrates the complexity of the role of a protein during the course of infection. If administered prior to infection, AnxA1 plays an immunomodulatory role that is beneficial to the host, expanding the AM pool via GM-CSF production and protecting animals against IAV, while its presence during the course of an infection facilitates viral replication in different phases of the viral life cycle [85,92,117]. Importantly, regardless of decreased viral loads in the lungs and trachea, the study of Schloer and colleagues showed that the treatment with AnxA1 did not alter the levels of type I interferons (IFNs) at either transcription or protein levels, and did not alter the levels of IFN-induced genes, such as Irf7 and Mx2, demonstrating that the binding of AnxA1 to FPR2 led to the decreased viral titers through specific mechanisms other than the expression of interferon-induced genes [19].

Bist and co-workers demonstrated that AnxA1KO mice have decreased levels of CXCL-10 and IFN-β in the serum after in vivo administration of poly:IC, a ligand of the endosomic sensor toll-like receptor 3 (TLR-3). Alongside that, in human 293 T embryonic kidney cells transfected to overexpress AnxA1, enhanced IFN-β promoter activity was found [86]. This was attributed to the physical interaction between AnxA1 and TANK-binding kinase 1 (TBK-1) in the cytoplasm of the cells, independently of the FPR2 receptor. Interestingly, this same study showed that the activation of FPR2 by AnxA1 decreased IFN-β transcription, suggesting a dual role of AnxA1 in the modulation of IFN-β transcription: it stimulates its expression when in the cytoplasm and inhibits it once secreted by binding to FPR2. Altogether, these results suggest that the modulation of IFN-β expression by AnxA1 may vary according to the cell type and the stimulus used. In fact, a bilateral relationship may exist between the FPR2 receptor and the signaling pathways related to IFN-β production, as Ampomah and colleagues showed that type I IFNs signaling mediates the up-regulation of FPR2 via IRF-3 and STAT-3 phosphorylation [87].

The use of antagonists of the FPR2 receptor during IAV infection was explored in different studies. WRW4 and BOC2 are commercially available antagonists of FPR2 that were tested in vivo during IAV infection and shown to protect mice from lethality associated to infection [85,118]. In short, AnxA1 and FPR2 receptors favor IAV replication through different mechanisms, including: (1) viral particles being adsorbed onto AnxA1 molecules and internalized via the FPR2 receptor; (2) downstream phosphorylation of ERK 42/44 occurring via the AnxA1/FPR2 axis, which in turn favors IAV replication; (3) AnxA1 increasing the import of NS1 IAV protein to the nucleus of the cells; and (4) FPR2 activation by AnxA1 enhancing the endosomal export of IAV particles, as the use of FPR2 monoclonal antibody or WRW4 inhibitor blocks the viral particles into the endosomes [91,92]. AnxA1 levels were increased in nasal swabs from Influenza-A-infected patients when compared to healthy controls, presumably as a means to try to control excessive inflammation [91]. Altogether, these results demonstrate the deleterious role of the AnxA1/FPR2 axis during IAV infection in mouse models. Therefore, in the context of influenza, inhibitors of FPR2 may hold great potential to treat IAV infection during possible outbreaks.

### 3.3. Annexin A1 during Non-Respiratory Viral Infections

#### 3.3.1. Arbovirus

Arbovirus infections, such as Chikungunya (CHIKV), Dengue (DENV) and Zika (ZIKV), can cause fever, rash, muscle, joint and bone pain. Although clinically indistinguishable at the very acute phases of the disease, there are certain features which are unique to each virus: CHIKV infection is characterized by disabling polyarthralgia [119], DENV infection can lead to hemorrhage [120], and ZIKV infection is associated with the condition of microcephaly [121].

Overall, the AnxA1 pathway appears to be beneficial for controlling excessive inflammation and hypernocyception in arboviral diseases, without affecting the ability of the host to fight infection [24,26]. Short-term systemic glucocorticoids have been successfully used to treat chronic rheumatic manifestations caused by CHIKV. As outlined above, AnxA1 is produced in response to glucocorticoids [122], and reduced AnxA1 levels have recently been associated with CHIKV-induced arthritis and may be a biomarker for determining the severity of the disease [123].

Interestingly, the AnxA1 mimetic peptide, Ac2-26, has been shown to effectively reduce inflammation and edema, as well as provide an analgesic effect via the activation of the FPR2 receptor in CHIKV infection. This suggests that targeting the AnxA1/FPR2 pathway might be beneficial in managing Chikungunya fever (CHIKF) and its associated arthritis [26,124].

In a recent study, Costa and co-workers found that DENV-infected patients had significantly lower levels of AnxA1 in their plasma compared to healthy controls. Furthermore, lower AnxA1 levels were associated with more severe forms of dengue infection. These findings suggest that AnxA1 could be a useful biomarker for predicting the severity of dengue infection, and could potentially be used to guide treatment [24]. Conversely, the treatment of WT mice with attenuated Ac2-26 resulted in thrombocytopenia, hemoconcentration and reduced plasma levels of MCPT-1 and CCL2 compared to the untreated group [24].

Like the DENV study, Molás and colleagues have demonstrated a correlation between reduced levels of AnxA1 in placental tissue and increased inflammation in pregnant women with ZIKV infection, suggesting that AnxA1 may play a role in the severity of ZIKV-associated complications [93]. ZIKV infection during pregnancy can cause severe complications, such as microcephaly, placental dysfunction and reduced fetal weight [121].

Collectively, these studies suggest that the AnxA1/FPR2 axis could be a potential target for host-directed therapies for arbovirus-induced infections, such as CHIKV, DENV, and ZIKV, due to its protective role in these diseases.

#### 3.3.2. Hepatitis C Virus (HCV)

Persistent HCV infection is a major global health concern as it can cause cirrhosis, chronic hepatitis and hepatocellular carcinoma, and can lead to severe health complications if left untreated [125]. Many of these problems have been clearly curtailed by the availability of very effective anti-viral drugs. Sejima and colleagues found that during long-term HCV replication, the expression of AnxA1 was irreversibly downregulated, suggesting that AnxA1 is a negative regulator of HCV susceptibility in human hepatocytes [126]. Further, the analysis of mRNA expression has demonstrated decreased levels of AnxA1 in D7 cells, which are highly permissive for HCV [94]. Of note, AnxA1 was found to block HCV RNA replication without interfering with viral entry in human hepatocytes [94]. Whether the use of AnxA1-based therapies will be useful in drug-resistant patients deserves further investigation.

#### 3.3.3. Human Papillomavirus (HPV)

Human papillomavirus (HPV) is the most common cause of cervical cancers [127]. Dysregulation of cell cycle control, including the inhibition or mutations in p53, are related to tumorigenesis [128]. The pro-apoptotic tumor suppressor p53 is a protein that plays a critical role in the control of division and cell survival [129]. However, p53 is inactivated by continuous E6 expression, an oncoprotein found in HPV, that forms a complex with a ubiquitin ligase, E6AP, which mediates p53 degradation [130,131,132]. In vitro and clinical studies suggest that AnxA1 is a biomarker in HPV-mediated carcinogenesis [96,132]. Indeed, an increase in AnxA1 expression was found in epithelial margins of tumors from patients with oropharyngeal cancer HPV positive and in samples from patients with penile carcinomas positive for high-risk HPV [95,96]. Another study revealed that E6 binds to E6AP to degrade p53 and upregulate AnxA1 in cells infected with HPV type 16, while in cells transfected with E6 DsiRNA, decreased levels of AnxA1 were found [132]. Moreover, AnxA1 silencing reduced HPV-transformed cell proliferation corroborating with the hypothesis that E6 could potentially activate AnxA1 expression to facilitate carcinogenesis [132]. Altogether, these studies indicate that infection with HPV enhances AnxA1 expression and this protein may play a role in HPV-mediated carcinogenesis.

#### 3.3.4. Simian Immunodeficiency Virus (SIV)

Perturbed inflammatory responses are often associated with immunodeficiency viruses. For example, the gut barrier dysfunction is observed in HIV patients, while progressive loss of CD4+ T cells occurs systemically [133,134]. Therefore, molecules that play a role during these perturbed inflammatory responses may represent interesting therapeutic targets. It was demonstrated that AnxA1 levels were decreased in the gut, and increased systemically during early infection of SIV in rhesus macaques. Conversely, AnxA1 levels were increased both in the gut and systemically during the chronic phase of infection in the animals [97]. Interestingly, these altered levels of AnxA1 in both compartments were correlated with perturbed levels of anti-inflammatory mediators, such as IL-10 and TGF-β, which were associated with immune dysregulation in HIV patients [97].

#### 3.3.5. Herpes Simplex Virus 1 (HSV-1)

The Herpes simplex virus 1 (HSV-1) is a neurotropic virus that causes latent infection in humans and can cause clinical symptoms ranging from painful sores to life-threatening neuroinflammation [135]. AnxA1 is expressed on the surface of cells that are known to be permissive for HSV-1 replication, including A549 and N18 cells (derived from mouse neuroblastoma) [20]. It was reported that the binding of AnxA1 to glycoprotein E (gE) on the HSV-1 envelope enhances virus binding and facilitates the infection of A549 or N18 cells. In addition, the FPR2 receptor antagonist, WRW4, was shown to reduce HSV-1 binding to the cell surface [20]. Furthermore, in HSV1 encephalitis in mice, the absence of AnxA1 or FPR2 was associated with reduced mortality and lower tissue viral loads compared to infected WT controls [20]. These findings suggest that the antagonism of the AnxA1/FPR2 axis could be a potential target for reducing the lethality of encephalitis caused by HSV-1.

#### 3.3.6. Human T-Lymphotropic Virus 1 (HTLV-1)

Studies suggest that AnxA1 plays an important role in the differentiation and function of T cells. AnxA1 appears to promote Th1 differentiation by binding to FPR2 and through the activation of the ERK signaling pathway. Furthermore, the absence of AnxA1, specifically in T cells, has been linked to increased inflammatory responses [136,137]. In this context, the role of AnxA1 in T cell diseases, such as HTLV-1 infection, is of great importance. Expression of AnxA1 in T-cell lineages infected with HTLV-1 in vitro may correlate with the stage of differentiation of these cells, and could thus be used as a disease biomarker [138]. Indeed, low levels of AnxA1 in the peripheral blood of HTLV-1 patients were observed, which may be related to an anti-inflammatory effect during infection. Furthermore, AnxA1 low levels were found to be inversely correlated with viral loads, although no correlation between these parameters was observed [102]. These results are still unclear, and further mechanistic studies may elucidate the exact role of AnxA1 during HTLV-1 infection.

#### 3.3.7. Human Immunodeficiency Virus (HIV)

In recent years, evidence has shown that HIV subtypes A and C, but not D subtype, can use FPR2 as a co-receptor [98,99]. The use of FPR2 in place of CCR5 may indicate the progression of HIV infection [100]. Mechanistically, a specific tyrosine residue at position aa 16 is essential for HIV to use FPR2 [101]. In vivo, the levels of AnxA1 increased during disease progression [139]. To date, no studies have explored the therapeutic potential of FPR2 antagonists, such as WRW4 or BOC, in in vivo or in vitro models of HIV/SIV infection.

#### 3.3.8. Measles/Reovirus

Interestingly, AnxA1 was demonstrated to be fundamental for pore expansion and syncytiogenesis of reptilian reovirus and measles virus. Using an RNA interference approach to silencing AnxA1 in vitro, Ciechonska and co-workers demonstrated that reovirus p14 protein physically interacts with AnxA1 to contribute to pore expansion in a calcium-dependent manner, as the chelation of intracellular calcium abrogated AnxA1-p14 interactions and syncytium formation [140]. AnxA1 interacts with the F and H proteins of the measles virus to promote pore expansion. These findings demonstrate how different viruses can hijack endogenous proteins to benefit their own replication.

#### 3.3.9. Foot and Mouth Disease Virus (FMDV)

It was demonstrated that the levels of AnxA1 increased in a time-dependent manner in vitro during different viral infections, such as HSV, Sendai virus, Seneca valley virus and FMDV. Interestingly, two bands of AnxA1 in a Western blot could be observed sixteen hours post-infection with FMDV, although viral proteins known to cleave host proteins, such as FMDV 3C and L proteases, did not cause the cleavage of AnxA1 [141], suggesting that the cleavage of AnxA1 occurs through other mechanisms, at least in this in vitro system. It is noteworthy that the FMDV 3A protein interacts physically with AnxA1, preventing it from binding to TBK-1, which is a fundamental step of AnxA1 to increase the expression of IFN-β [86,141].

## 4. Concluding Remarks

The AnxA1/FPR2 axis is an important counter-regulatory branch of the immune system, triggered by pro-resolving activities and the resolution of inflammation. Given its crucial role in controlling inflammation, harnessing this pathway has been exploited pharmacologically to treat inflammatory diseases. Viral diseases are often related to dysregulated inflammatory responses and, therefore, one would assume that AnxA1 would be an interesting therapeutic strategy. While this is potentially true for some viral infections, including arbovirus diseases, other viruses have evolutionarily developed strategies to hijack the AnxA1/FPR2 axis, to infect and replicate more effectively. An interesting remark, based on the current literature on AnxA1 and viral disease, is that viruses that have co-evolved closely with humans, such as seasonal influenza virus and HSV-1, were shown to exploit the AnxA1 pathway for replication. In contrast, emerging viruses or viruses that adapted to humans more recently, usually trigger a more prominent inflammatory response and the infected individuals might benefit most from AnxA1 increasing pharmacological strategies.

Here, we discussed the role of AnxA1 during different viral infections and this is briefly summarized in Figure 1 and Figure 2. The therapeutic potential of AnxA1, Ac2-26 or FPR2 antagonists may vary according to the pathogen, timing of administration and relevance for exacerbated inflammation in disease pathogenesis. Studies in pre-clinical models have been instrumental to uncover AnxA1 protective or detrimental effects in the context of viral diseases. Further studies will be necessary if we are to translate the current knowledge on the biology of AnxA1 to antiviral or anti-inflammatory treatments in humans.

## Figures and Tables

**Figure 1 cells-12-01131-f001:**
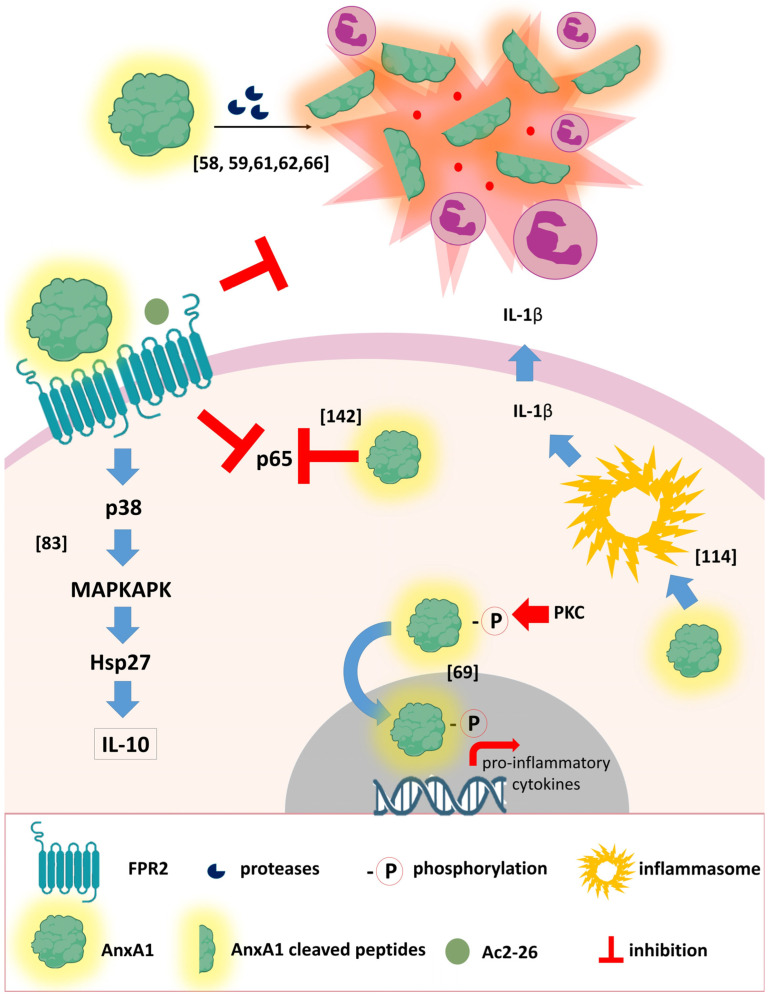
The intricate role of AnxA1 in different signaling pathways. As a pleiotropic molecule, AnxA1 can stimulate or inhibit different signaling pathways in the cytoplasm of the cells, leading to anti- or proinflammatory effects. For instance, it was demonstrated that the binding of AnxA1 and Ac2-26 occurs preferentially to FPR2 homodimers, activating signaling pathways dependent on p38 and Hsp27, leading to IL-10 production. Conversely, if phosphorylated by PKC at Serine 27 and translocated to the nucleus, AnxA1 can stimulate the transcription of proinflammatory cytokines. Additionally, different studies indicate that the peptides obtained from the cleavage of AnxA1 are certainly related to inflammation in the tissue and neutrophil infiltrate. AnxA1 can also activate the NLRP3 inflammasome to induce IL-1β release. Moreover, the binding of AnxA1 to the FPR2 receptor leads to the inhibition of NF-κB subunit p65, whilst solely in the cytoplasm, AnxA1 can inhibit this same protein. Evidence for each of these phenomena are referenced within the figure [58,59,61,62,66,69,83,124,142].

**Figure 2 cells-12-01131-f002:**
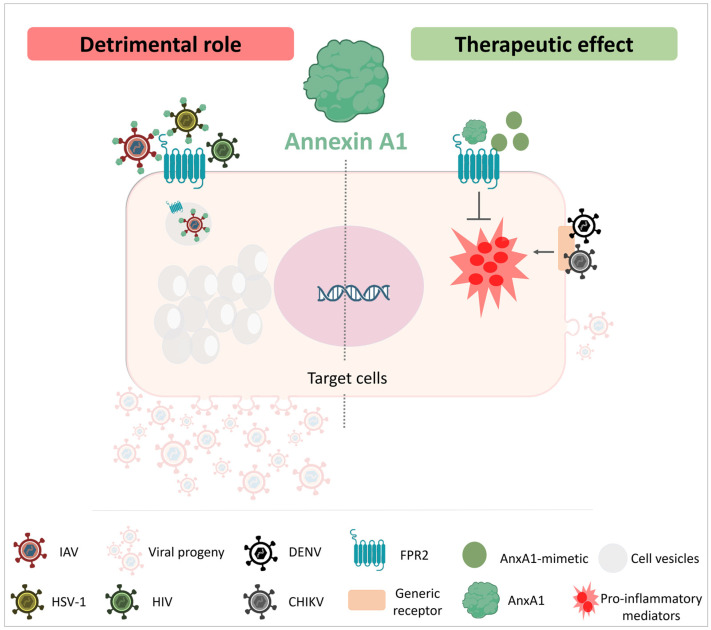
The multifaceted role of Annexin A1 in viral infections. In the left part of the figure, viral infections in which endogenous AnxA1 plays a deleterious role are represented, such as increasing virus endocytosis and intervening with different mechanisms in the cell cytoplasm that favor viral replication. Viruses such as IAV and HSV-1 use FPR2 as a co-receptor, with AnxA1 adsorbed in their viral particles. Conversely, HIV did not depend on AnxA1 to use FPR2 as a co-receptor. In the right part of the panel, viral infections in which AnxA1 or its mimetic peptides play a beneficial role are represented, such as for CHIVK and DENV, decreasing the production of pro-inflammatory mediators and exuberant inflammation in a general way, without altering the capacity of the host to deal with viral replication.

**Table 1 cells-12-01131-t001:** Clinical studies and experimental animal models evaluating the role of AnxA1/FPR2 in viral infections.

	**(a)** **Respiratory Infections**		
Virus	In Vivo (Model); In Vitro (Cell Type)	Comment	Reference
SARS-CoV-2 (COVID-19)	Case-control study with analysis of clinic-based blood samples	Lower AnxA1 levels were found in the severe/critical disease group compared with the control and moderate disease groups	[89]
	Prospective cohort study with analysis of clinic-based blood samples	Elevated levels of AnxA1 were observed in cases of moderate and severe disease	[90]
Influenza A virus (IAV)	Human epithelial cell line (A549) or MDCK cells; C57BL/6 (WT mice)	Activation of FPR2 stimulated viral replication through an ERK-dependent pathway, resulting in reduced survival in mice	[85]
	AnxA1-KO and WT mice; A549 cells	AnxA1 increased viral replication, affected virus binding, and promoted the trafficking of endosomal viruses to the nucleus	[91]
	A549 cells	The blocking of FPR2 signaling inhibited viral replication and interfered with the endosomal trafficking of IAV	[92]
	C57BL/6 (WT) mice; samples: lungs of mice	The activation of the AnxA1/FPR2 pathway demonstrated beneficial effects for the host, improving survival, inhibiting viral replication and expanding alveolar macrophages	[19]
	A549 cells	AnxA1 plays a regulatory role in RIG-I signaling and induces apoptotic cell death upon infection	[21]
	**(b)** **Non-respiratory infections**		
**Virus**	**In vivo (model); in vitro (cell type)**	**Comment**	**Reference**
Chikungunya (CHIKV)	AnxA1-KO; FPR2-KO and WT mice	*The activation of the AnxA1/FPR2 pathway demonstrated beneficial effects for the host by reducing the inflammatory response*	[26]
Dengue virus (DENV)	AnxA1-KO; FPR2-KO; A129, and WT mice	*The activation of the AnxA1/FPR2 pathway demonstrated beneficial effects for the host*	[24]
Zika virus (ZIKV)	Cross-sectional study with analysis of clinic-based samples: placental fragments of pregnant women with suspected Zika virus infection	*AnxA1 is involved in modulating inflammation in response to ZIKV infection*	[93]
Hepatitis C virus (HCV)	Human hepatoma cell line Li23 and Li23-derived D7 cells	*AnxA1 negatively regulates the step of viral RNA replication, but does not regulate viral entry*	[94]
Human papillomavirus (HPV)	Analysis of clinic-based samples: tissue sections from penile squamous cell carcinoma	*AnxA1 is overexpressed in high-risk HPV-positive penile carcinoma patients*	[95]
	Analysis of clinic-based samples: tumor and adjacent mucosa from patients with squamous cell carcinoma of the oropharynx	*Increased expression of AnxA1 in HPV^+^ samples suggests that the protein is involved in the early stages of HPV-driven carcinogenesis*	[96]
Simian immunodeficiency virus (SIV)	Rhesus macaques	*During early stages of infection, AnxA1 expression decreased in the gut and increased in the blood, while during chronic infection, AnxA1 expression increased in both compartments*	[97]
Human immunodefiency virus (HIV)	NP-2 cells; C8166 cells; T CD4+ cells obtained from human donors	*HIV strains used FPR2 efficiently as a co-receptor; 22 envelope proteins can bind to FPR2*	[98,99,100,101]
Herpes simplex virus 1 (HSV-1)	AnxA1-KO and WT mice; A549 cells	*Similar to IAV, viral particles of HSV-1 hijack the AnxA1/FPR2 pathway to increase endocytosis by the cells* via *FPR2 receptor*	[20]
Human T-lymphotropic virus 1 (HTLV-1)	Analysis of clinic-based samples: blood	*An imbalance in the expression of ANXA1 may contribute to the development of chronic neurodegenerative diseases caused by HTLV-1*	[102]

Abbreviations: AnxA1, annexin A1; FPR, formylated peptide receptor; WT, wildtype; KO, knockout.

## Data Availability

Not applicable.

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
