# Peer review of "The Multifaceted Role of Annexin A1 in Viral Infections"

_cells, 2023, doi:10.3390/cells12081131_

Round 1

Reviewer 1 Report

The manuscript titled "The multifaceted role of Annexin A1 in viral infections" examines the role of Annexin A1 (AnxA1), an endogenous pro-resolving protein, in regulating inflammation during viral infections. AnxA1 aids in controlling inflammation by activating signaling pathways that contribute to the termination of the response, pathogen clearance, and restoration of tissue homeostasis. Employing AnxA1 as a therapeutic strategy may help manage the severity of clinical presentations of viral infections. The review offers a comprehensive analysis of AnxA1's role in viral infections, encompassing both pre-clinical and clinical studies, and delves into the therapeutic potential of AnxA1 and AnxA1 mimetics for treating viral infections.

  1. In either Section 2 or 3, the role of Anxa1 in pro-inflammatory and anti-inflammatory processes should be clearly illustrated. To effectively demonstrate the role of Anxa1 in inflammation, it would be beneficial to use a figure depicting the various signaling pathways involved.
  2. The authors have discussed the role of Anxa1 in a range of virus types, such as COVID, IAV, Arbovirus, etc., categorized by respiratory and non-respiratory. Although this part is well-written, the detailed underlying molecular mechanisms that explain how Anxa1 functions in viral infections have not been thoroughly reviewed. The authors might consider adding a summary in the discussion section or elsewhere that elaborates on how Anxa1 interacts with different genome structure types, such as DNA/RNA viruses and single or double-stranded viruses.

Author Response

We thank the valuable comments raised by the reviewer 1. We have worked on the comments and updated them in the resubmitted version of the manuscript. The modifications are as follow: 

  1. In either Section 2 or 3, the role of Anxa1 in pro-inflammatory and anti-inflammatory processes should be clearly illustrated. To effectively demonstrate the role of Anxa1 in inflammation, it would be beneficial to use a figure depicting the various signaling pathways involved.

Response: Regarding that, we have included a figure between the sections 3.1 and 3.2 to inform the readers about this interesting aspect of Annexin A1. The interaction between Annexin A1 and different proteins of the cytoplasm is not fully understood and requires more evidence in future studies. Therefore, we included the references for each of the described role of Annexin A1 within the figure.

  1. The authors have discussed the role of Anxa1 in a range of virus types, such as COVID, IAV, Arbovirus, etc., categorized by respiratory and non-respiratory. Although this part is well-written, the detailed underlying molecular mechanisms that explain how Anxa1 functions in viral infections have not been thoroughly reviewed. The authors might consider adding a summary in the discussion section or elsewhere that elaborates on how Anxa1 interacts with different genome structure types, such as DNA/RNA viruses and single or double-stranded viruses.

Response: We appreciate your interest in the potential role of AnxA1 in viral infections. However, as of the current literature review, there are no studies exploring the interaction between AnxA1 and viral genome structures. While AnxA1 has been reported to facilitate the endocytosis of certain viruses and modulate the transcription of IFN-β gene, there is no evidence to suggest a direct interaction with viral genome structures. Therefore, unfortunately, we are unable to include a topic on this issue in the manuscript. We believe that our review can still summarize and contribute to the understanding of the multifaceted role of AnxA1 in various viral infections.

Finally, we appreciate your thoughtful feedback and thank you for your time and effort in reviewing our manuscript.

Reviewer 2 Report

This is a very comprehensive review of roles of Annexin A1 in viral diseases. The complexity of distinct functions of Annexin A1 is well described. This review will be of great interest for researchers in annexin biology as well as for researchers in the field of viral diseases.
I have nothing to add.

Author Response

Thank you for your valuable feedback on our manuscript.